# Coexisting Nodular Sclerosis Hodgkin Lymphoma and Kimura’s Disease: A Case Report and Literature Review

**DOI:** 10.3390/ijms24087666

**Published:** 2023-04-21

**Authors:** Chih-Chun Lee, Sing-Ya Chang, Wen-Chieh Teng, Chih-Ju Wu, Chi-Hung Liu, Szu-Wei Huang, Chiao-En Wu, Kuang-Hui Yu, Tien-Ming Chan

**Affiliations:** 1Department of Medical Education, Chang Gung Memorial Hospital, Keelung Branch, Keelung 20401, Taiwan; 2Department of Medical Education, Chang Gung Memorial Hospital, Linkou Branch, Taoyuan 33305, Taiwan; 3Division of Rheumatology, Allergy, and Immunology at Chang Gung Memorial Hospital, Linkou Branch, and Chang Gung University College of Medicine, Taoyuan 33305, Taiwan; 4Division of Hematology-Oncology, Department of Internal Medicine, Chang Gung Memorial Hospital, Linkou Branch, Chang Gung University College of Medicine, Taoyuan 33305, Taiwan

**Keywords:** Kimura’s disease, Hodgkin lymphoma, case report, lymphoma, nodular sclerosis Hodgkin lymphoma

## Abstract

Kimura’s disease (KD) is a rare lymphoproliferative fibroinflammatory disorder that commonly affects the subcutaneous tissue and lymph nodes of the head and neck. The condition is a reactive process involving T helper type 2 cytokines. Concurrent malignancies have not been described. Differential diagnosis with lymphoma can be challenging without tissue biopsy. Here, we present the first reported case of coexisting KD and eosinophilic nodular sclerosis Hodgkin lymphoma of the right cervical lymphatics in a 72-year-old Taiwanese man.

## 1. Introduction

Kimura’s disease (KD) is a rare, chronic lymphoproliferative fibroinflammatory disorder associated with type 1 hypersensitivity and T helper (Th) type 2 cytokines [1]. The entity was first described by Kimm and Szeto [2] in 1937, and later characterized by Kimura et al. [3] in 1948. The patients affected are classically young to middle-aged males of Asian descent. The clinical presentation typically involves one to multiple painless, slow-growing subcutaneous nodules of the head and neck. Cervical lymphadenopathy is common and can be the only presenting symptom. Less commonly is the involvement of the axillary or inguinal lymph nodes. Further workup characteristically reveals a disease process affecting the subcutaneous tissue, lymph nodes, and salivary glands. Peripheral blood eosinophilia and elevated serum immunoglobulin E (IgE) levels are variably present. Patients with KD can have comorbid Th2 and/or eosinophilic conditions, including atopy and eosinophilic infiltration of various tissues or organs. Visceral involvement constitutes the most significant morbidity and can develop years after the onset of soft tissue disease. The kidneys are the most frequently involved. KD is believed to be benign. Concurrent malignancies have not been described.

## 2. Case Report

A 72-year-old Taiwanese man was referred to our hematology clinic after a right cervical lymph node biopsy that was suspicious of lymphoid malignancies. The patient reported a one-month history of multiple swollen neck masses and an unintentional four kilogram body weight loss within three months. No obvious prodrome was observed before the onset of lymphadenopathy. The swollen lymph nodes were non-tender, with no associated fever or night sweats. According to the patient, he was healthy before the current episode. He had not been exposed to any immunosuppressive agents or vaccinations prior to the disease onset.

The patient’s past medical and personal histories were unremarkable with no documentation of recent medications in his medical chart. He had no family history of allergies, T helper (Th) type 2 conditions, or lymphoid neoplasms. He was retired with no recent travel or contact history. A physical examination revealed numerous one to four centimeters. Laboratory examination showed mildly elevated eosinophils (6.0%, reference range: 0%–5%) and monocytes (13.0%, reference range: 0%–12%), presence of atypical lymphocytes (0.5%), and elevated level of β2-microglobulin (3653.0 ng/mL, reference range: ≤2366 ng/mL). Plasma Epstein–Barr virus (EBV) DNA titre was undetectable. Screening for HBV and HCV infections also showed negative results.

Histopathological examination of the resected lymph node showed follicular hyperplasia with well-formed germinal centers, interfollicular dense eosinophilic infiltrate with eosinophilic microabscesses, vascular proliferation, and prominent interstitial fibrosis, all of which are characteristics of KD (Figure 1). No signs of malignancy were seen. Due to the patient’s unusual demographic feature (old age) and suspicious history (recent body weight loss), the initial biopsied specimen from his previous healthcare provider was obtained. The sample was then re-examined by our specialists with additional immunochemical studies. The lesion was characterized by a thick fibrous capsule, focal nodular sclerosis, near-total nodal structure effacement, and increased vascularity. Scattered Hodgkin-like and Reed–Sternberg-like cells were identified in a background of small lymphocytes, plasma cells, neutrophils, histiocytes, and an excess of eosinophils (Figure 2). Immunohistochemical studies (Figure 3) further characterized the Reed–Sternberg-like cells as CD3 (−), CD20 (+), CD30 (+), CD15 (−), and PAX5 (+/−), consistent with nodular sclerosis classical Hodgkin lymphoma (NSHL). To differentiate the lesion from non-Hodgkin lymphomas, additional staining of BCL2, BCL6, GCET1, MUM-1, OCT-2, CD5, and FOXP1 was performed. The results were all negative. Specifically, the absence of OCT-2 differentiated the lesion from common mimickers, such as grey zone lymphoma or EBV-positive diffuse large B-cell lymphoma. Staining of CD10, CXCL13, and CD21 also showed negative results, ruling out the angioimmunoblastic T-cell lymphoma. The series of workup confirmed the pathological diagnosis of eosinophilic NSHL with coexisting KD.

With the diagnosis of Hodgkin lymphoma, bone marrow examination and whole-body positron emission tomography-computed tomography (PET-CT) were subsequently arranged for staging and treatment planning. Interestingly, bone marrow eosinophilia (6.5% of total neutrophil count with a bone marrow cellularity of 40%; reference range <5.0%) was noted by the hematologists. Whole-body PET-CT revealed malignant involvement of the right cervical level III and V and supraclavicular lymph nodes (Figure 4), consistent with the clinical diagnosis of stage I Hodgkin lymphoma according to the Ann Arbor staging system.

Under the impression of stage I NSHL involving the right neck lymph nodes with old age and an Eastern Cooperative Oncology Group performance score of 0, the patient received four cycles of chemotherapy with the ABVD regimen (doxorubicin [25 mg/m^2^], bleomycin [10 mg/m^2^], vinblastine [6 mg/m^2^], and dacarbazine [375 mg/m^2^]), followed by consolidative local radiation therapy (RT) (3060 centigrays [cGy] in 17 fractions) of the right neck lymphatics. The site of the completely resected KD lesion and its surrounding area were included in the clinical target volume of the RT. The patient tolerated and responded well to the surgery, chemotherapy, and RT without the occurrence of any clinically significant adverse event. Subsequent laboratory and imaging monitoring performed at 6-month intervals showed complete remission of both NSHL and KD. Neither NSHL nor KD recurred at 9 years.

## 3. Discussion

To the best of our knowledge, our report represents the first documented case of concurrent lymphoma and KD occurring in close proximity in the same patient. The histopathological diagnoses were unequivocal for both lesions, representing classical KD and eosinophilic NSHL, respectively.

The etiology of KD is incompletely understood. It is believed to be a reactive process. This is corroborated by the histopathological features exhibiting exuberant follicular hyperplasia, inter- and intra-follicular dense eosinophilic and lymphoplasmacytic infiltration, eosinophilic microabscesses (sometimes with Charcot-Leyden crystals), proliferation of small blood vessels, and significant stromal fibrosis. Proliferation of postcapillary venules is especially prominent. Architectural distortion of germinal centers is uncharacteristic and should prompt further evaluation for mimickers such as immunoglobulin G4-related disease [4]. However, replacement of normal germinal centers with IgE or eosinophilic deposits, necrosis, and Warthin–Finkeldey-type giant cells (polykaryocytes) may be present [4,5]. In some patients, there is accompanying inflammatory infiltration of nerve fibers, which might explain the clinical symptoms of skin irritation and pruritus [6].

The development of KD is presumed to be caused by immunologic dysregulation. Triggering factors, including allergies, infections, endocrine disorders, and autoimmunity, have been proposed in association with a phenotype involving both IgE-mediated type 1 hypersensitivity and Th2 cytokines [7,8,9,10]. The Th2 signatures were supported by the observation of elevated interleukin (IL)-4, IL-5, and IL-13 mRNA expression in the peripheral blood mononuclear cells of patients with KD [11]. Local infiltration of IL-4+ and IL-5+ mast cells and T cells has also been demonstrated [12]. Furthermore, the levels of eotaxin+ and C–C motif ligand 5 (CCL5; also known as regulated upon activation, normal T cell expressed, and presumably secreted [RANTES])+ mast cells, T cells, and eosinophils were also elevated in KD lesions [12]. We recently published an updated pathogenesis of KD incorporating the concept of “allergic fibrosis” [1]. This was extrapolated from an innovative comparative study of follicular T helper cells participating in KD and IgG4-related disease, in which the researchers differentiated between “allergic fibrosis” (Th2-driven) and “inflammatory fibrosis” (driven by cytotoxic T cells) based on the differential infiltrating follicular T helper cell subtypes [13]. Further transcriptomic studies would hopefully help elucidate the pathophysiological process of KD.

Differentiating between KD and lymphomas can be challenging. An early distinction between the two is crucial, as delayed diagnosis of lymphoid neoplasms can adversely affect the disease prognosis. Constitutional symptoms and features of local invasion are generally unusual in patients with KD. However, some exceptions may exist. One patient with KD-associated generalized lymphadenopathy was reported to suffer from compressive symptoms causing dysphagia [14], mimicking malignant lymphomas. Previous studies have attempted to distinguish the two processes via imaging studies. In a case series of 13 patients, it was suggested that given the highly variable computed tomography (CT) imaging characteristics, KD lesions should be viewed on a continuum from relatively well-defined, homogeneously enhancing acute lesions (type 1) to ill-defined, poorly enhancing chronic lesions (type 2) [15]. Another study of seven patients characterized subcutaneous head and neck KD lesions as exhibiting gradual upward enhancement on dynamic contrast-enhanced magnetic resonance imaging (MRI) and a lack of high intensity on diffusion-weighted images [16]. Nevertheless, imaging studies, including CT, MRI, and PET, were often inadept in guiding clinical decisions in practice, as many KD lesions with active inflammation demonstrate features virtually indistinguishable from that expected with lymphomas [17]. Wang et al. presented a case of KD-associated generalized lymphadenopathy with 18F-fluorodeoxyglucose-PET imaging features closely resembling lymphomas or nodal metastases [18]. Recently, a rare case of inguinal KD was reported, in which both MRI and 18F-fluorodeoxyglucose-PET-CT imaging studies suggested malignancy [19]. Alternatively, the possible evolution of KD lesions into peripheral T-cell lymphomas has been described in earlier reports [20,21]. However, evidence on the presence of clonal T-cell expansion in KD is scarce and conflicted [22,23], and the validity of malignant transformation remains inconclusive. The bottom line is that KD and lymphomas can be virtually indistinguishable clinically. Tissue biopsy remains the gold standard, especially when there is clinical suspicion of more aggressive processes. Moreover, a lower threshold for tissue biopsy should probably be considered for older and even younger patients with risk factors for lymphoid malignancy development. Our patient’s post-operative PET-CT study indicated malignant involvement of the right cervical levels III and V and supraclavicular lymph nodes. Given the existing evidence, the nature of these atypical lymph nodes could not be definitely determined without tissue proof. However, they were prudently considered an NSHL involvement. Moreover, the contiguous pattern of the affected lymph nodes was compatible with Hodgkin lymphoma.

Historically, the treatment of KD comprises of surgical resection, RT, and medical immunomodulation, including corticosteroids, cyclosporine, mycophenolate mofetil or mycophenolic acid, leflunomide, mepolizumab, and tacrolimus. However, the recurrence is high with either surgical resection or RT alone [24]. Currently, surgical resection remains the mainstay for treating local KD lesions. RT or immunosuppressive agents are considered adjuvants as part of the salvage therapy or prevention of recurrence. A recently proposed treatment algorithm stratified patients according to lesion size, disease duration, and laboratory results [25]. The researchers found that patients with larger tumors (>3 cm in diameter), longer disease duration (above five years), higher peripheral eosinophil counts (>20%), or higher levels of serum IgE (more than 10,000 international units [IU]/milliliter) were more prone to post-operative relapses or KD recurrences [25]. For our patient, the KD lesion was resected. The subsequent RT for the NSHL, with a clinical target volume comprising the right neck lymphatics, aided KD control. There was no recurrence.

Initial assessment and subsequent monitoring of renal involvement constitute an important aspect of clinical care for patients with KD. Unfortunately, despite the established association, routine screening and long-term follow-up for renal involvement are often overlooked and under-emphasized in clinical practice [1]. In our case, the initial urinalysis was unremarkable. Subsequent laboratory monitoring consistently showed normal estimated glomerular filtration rates. However, renal involvement in KD commonly manifests as asymptomatic proteinuria, microscopic hematuria, and hypertension, which are not readily detectable via blood work alone [26,27]. The onset of renal disease can be insidious or fulminant. Temporary hemodialysis is sometimes required. The histologic phenotypes range from minimal change disease, membranous nephropathy, focal segmental glomerulosclerosis, mesangial proliferative glomerulonephritis, and IgA nephropathy to acute tubular injury [26,27]. Interestingly, the myriad of renal manifestations of KD are unrestricted to eosinophilic processes, suggesting a currently unknown additional pathological mechanism. Most patients respond well to systemic corticosteroids. However, progression to end-stage renal disease is common [26,27]. Therefore, it is worth noting that in addition to blood work, routine urinalysis and blood pressure monitoring are crucial components of care for patients with KD, as renal involvement is an important cause of morbidity in this relatively benign condition.

Classical Hodgkin lymphoma is a distinctive immunohistological unit characterized by Hodgkin Reed–Sternberg cells with overexpression of programmed death-ligand 1 (PD-L1), a surface ligand crucial for tumor immune evasion [28]. Classical Hodgkin lymphoma has been described as a highly fatal disease in the past, leading to deaths of young patients within months of onset. With the advent of modern therapy, first with RT and later with the addition of the ABVD regimen, the prognosis of the early-stage disease is now excellent [29]. Compared with Western countries, the phenotype is relatively rare in the Asian population, with non-Hodgkin lymphomas representing the majority. It was reported that only 6.09% of all lymphoid neoplasms in Taiwan were Hodgkin lymphomas [30], with an incidence of 1–2/100,000 [31]. NSHL constitutes the most classical Hodgkin lymphoma, with a higher frequency in younger patients. An inflammatory infiltrate rich in eosinophils is not rare in NSHL [32]. Eosinophils possess the biological role of upregulating the expression of PD-L1 via NLR Family CARD Domain Containing 4 (NLRC4) inflammasome-mediated action and promoting the secretion of a proliferation-inducing ligand (APRIL) [33,34] (Appendix A). Given the significant eosinophilic infiltrates in the patient’s lymphoma tissue and his predilection for generalized eosinophilia, a shared pathophysiological link between the two coexisting conditions is plausible. Both KD and NSHL are uncommon diseases, and the significance of their coexistence in close proximity in the same patient remains unelucidated. Likewise, whether therapeutic manipulation of eosinophils can be of use in our patient’s predicament remains unanswered. In practice, a review of the system focusing on comorbid Th2 and eosinophilic conditions is warranted in all patients with KD, as these patients might benefit from additional systemic immunotherapy. Antagonism of the Th2 axis and/or eosinophils may be justified for those who show a generalized tendency to associated Th2-skewed comorbidities. In our case, the patient showed excellent response to conventional management of both KD and classical Hodgkin lymphoma without requiring additional immunomodulation.

## 4. Conclusions

We presented the first documented case of pathologically proven coexisting KD and eosinophilic NSHL in the same patient. Additional workup is warranted in older patients with KD, especially for those with atypical presentation.

## Figures and Tables

**Figure 1 ijms-24-07666-f001:**
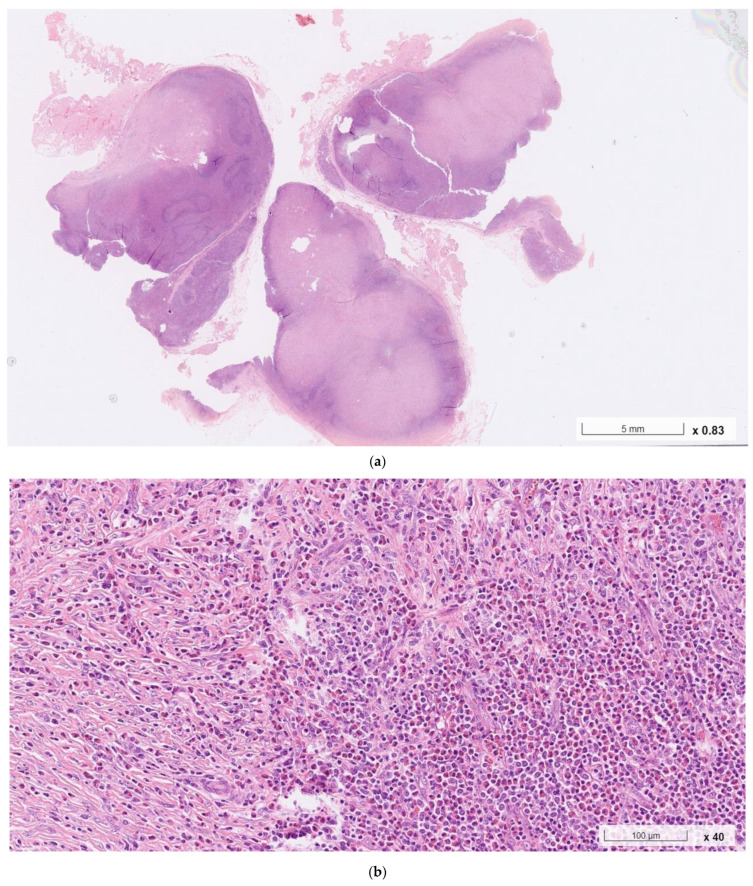
Histopathological examination of an enlarged right cervical lymph node; collected via complete surgical resection at our institution. (**a**–**c**): Hematoxylin–eosin staining showed the follicular hyperplasia of the lymphoid tissue with well-formed germinal centers, interfollicular dense eosinophilic infiltrates, eosinophilic microabscesses, vascular proliferation, and prominent interstitial fibrosis.

**Figure 2 ijms-24-07666-f002:**
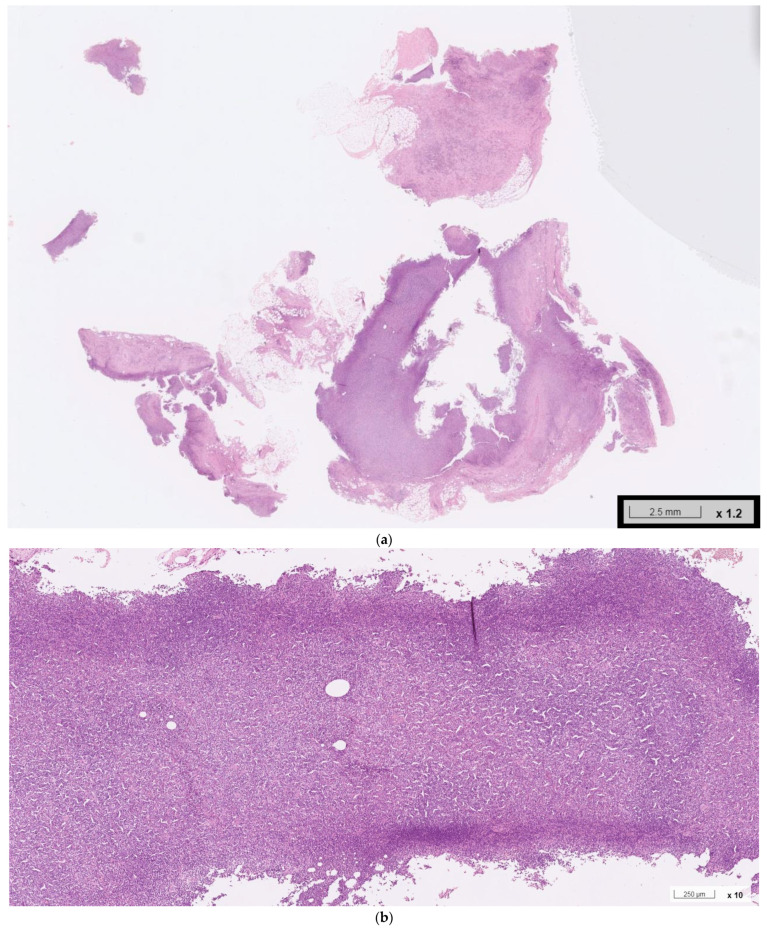
Histopathological examination of enlarged right cervical lymph node; obtained from his previous healthcare provider. (**a**–**c**): Hematoxylin–eosin staining demonstrated a thick fibrous capsule, focal nodular sclerosis, near-total nodal structure effacement, and increased vascularity. Scattered Hodgkin-like and Reed–Sternberg-like cells were seen in the background of small lymphocytes, plasma cells, neutrophils, histiocytes, and an excess of eosinophils.

**Figure 3 ijms-24-07666-f003:**
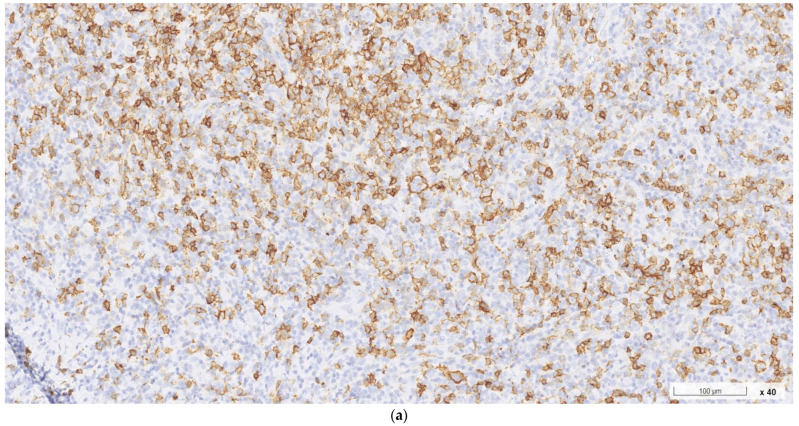
Immunohistochemical studies of an enlarged right cervical lymph node; obtained from his previous healthcare provider. (**a**–**c**): The Reed–Sternberg-like cells were positive for CD20, CD30, and PAX5, respectively.

**Figure 4 ijms-24-07666-f004:**
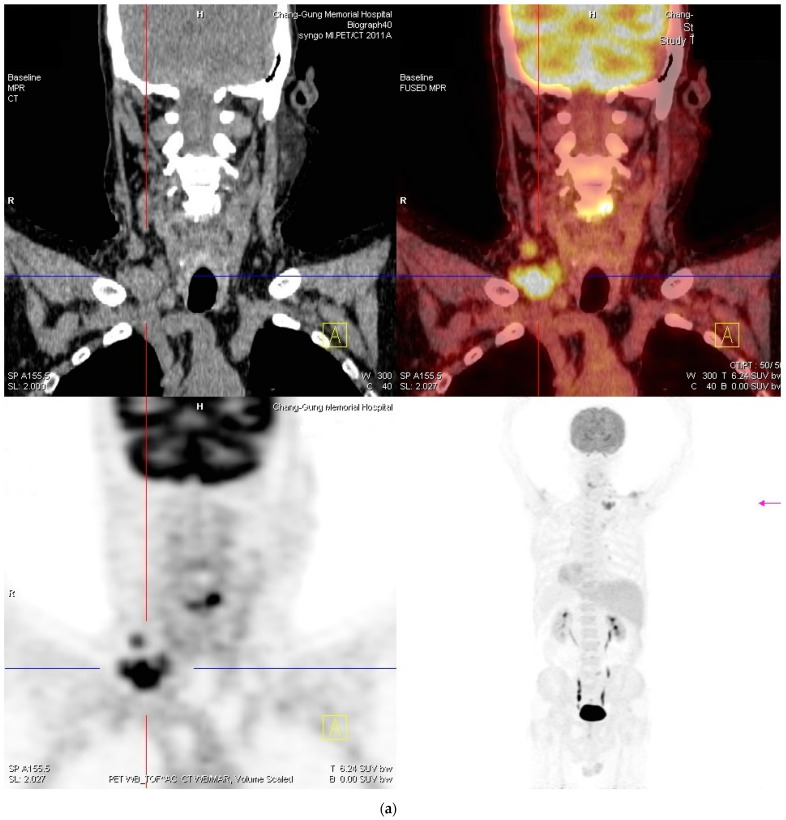
(**a**,**b**): Whole-body positron emission tomography-computed tomography revealed malignant involvement of the right cervical level III and V and supraclavicular lymph nodes. The uptake of radiation was elevated to standardized uptake value (SUV) 9.19, 3.79, and 10.15, respectively, corresponding to a score of 4 (high probability of malignancy).

## Data Availability

Not applicable.

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
