# Peer review of "Coexisting Nodular Sclerosis Hodgkin Lymphoma and Kimura’s Disease: A Case Report and Literature Review"

_ijms, 2023, doi:10.3390/ijms24087666_

Round 1
Reviewer 1 Report
Authors report the first documented case of pathologically proven concurrent a rare lymphoproliferative fibroinflammatory disorder (Kimura’s disease) and nodular sclerosis Hodgkin lymphoma of the right cervical lymphatics in the 72-year-old patient. The case report is well written and interesting. I have only one suggestion to revise and more actually references.
Authors report the first documented case of pathologically proven concurrent a rare lymphoproliferative fibroinflammatory disorder (Kimura’s disease) and nodular sclerosis Hodgkin lymphoma of the right cervical lymphatics in the 72-year-old patient. The case report is well written and interesting. I have only one suggestion to revise and more actually references.
Author Response
Dear the reviewer,
Thank you for your positive reply.
On behalf of the authors, I hereby submit our revised manuscript titled “Coexisting Nodular Sclerosis Hodgkin Lymphoma and Kimura’s Disease: A Case Report and Literature Review” (Manuscript ID: ijms-2288632) to the International Journal of Molecular Sciences. We appreciate the thoughtful comments of the reviewers and have revised our case report and review according to your suggestions.
Thank you for considering our manuscript for publication in the International Journal of Molecular Sciences. Please feel free to contact us if there are any additional questions or suggestions.
Best regards,
Tien-Ming Chan
Corresponding Author
joymingo@gmail.com

Reviewer 2 Report
Interesting report on association od ìf Hodgkin lymphoma and Kimura's disease in an edlerly patient , thus triple rare situation as both HL and KD ussually occur in t a youinger popultion
The paper is well wiritten and both clinical anc+d histopathological characteristics are fully described. The language is clear and figure well shown and supported
Of course a review by an histopatologist and not only by a clinician is needed
Both discussion and conclusions should be shortened ansìd synthetized as the text is oftenn redundand
There is a little need of english language
Author Response

(The authors gave the same response as above.)

Reviewer 3 Report
Cannot access supplementary figure S1.
Materials and methods not described (e.g. antibodies used)
Minor linguistic corrections:
Line 16 and line 112: Concurrent instead of contemporary
Line 17: Here instead of Hereby
Line 24: Rare instead of uncommon
Line 117: "Florid"???
Line 122: "Workup"???
Author Response
Dear editors and reviewers,
On behalf of the authors, I hereby submit our revised manuscript titled “Coexisting Nodular Sclerosis Hodgkin Lymphoma and Kimura’s Disease: A Case Report and Literature Review” (Manuscript ID: ijms-2288632) to the International Journal of Molecular Sciences. We appreciate the thoughtful comments of the reviewers and have revised our case report and review according to your suggestions. The revisions are marked with “Track Changes” function of MS Word.
Specifically, the following changes have been made:
- The Conclusions has been significantly shortened with selective information incorporated into the Discussion. This was done at Reviewer 2’s request to avoid repetitively and redundancy of these two sections.
- Minor changes to the use of words have been done in accordance and in addition to Reviewer 3’s suggestions.
We appreciate the careful consideration of our manuscript by the reviewers, and we’d also like to include our responses to the reviewers’ comments:
- We appreciate the encouraging comments from Reviewer 1.
- We appreciate the encouraging comments from Reviewer 2. In response to the concerns regarding the length of the latter sections, as we have received instructions from the editorial office to broaden the scope of our discussion, we felt obligated to commit to a more comprehensive review of this rare disease with detailed explanations. To avoid redundancy, we have cut down the Conclusions and incorporate relevant information into the Discussion, as well as made changes to the sentences to be more concise.
- We appreciate the detailed suggestions from Reviewer 3. We have revised our manuscript accordingly. We will upload the supplemental material again along with the revised manuscript. As our case study was limited by the retrospective nature, we are unable to obtain detailed information regarding the process of the initial preparation of the specimens. However, we can provide the final histological results (including IHC) in the form of digitalized photos. We’d be happy to provide additional supplementary figures if it is deemed necessary by the editors and reviewers.
- We appreciate Reviewer 4’s comments. We have previously discussed with the editorial office with regard to the journal’s desired article type and content. We have since committed to provide a comprehensive overview of KD in addition to the case study.
Thank you for considering our manuscript for publication in the International Journal of Molecular Sciences. Please feel free to contact us if there are any additional questions or suggestions.
Best regards,
Tien-Ming Chan
Corresponding Author
joymingo@gmail.com

Reviewer 4 Report
The journal is focused on original research papers and reviews, not case reports. I would recommend the authours to submit the article in a joornal which is focused on case reports.
Author Response

(The authors gave the same response as above.)

Round 2
Reviewer 4 Report
Although this case report is certainly interesting, the aim and scope of the proposed journal lie primarily in basic and translational science. This type of manuscript is more apppropriate for the journals dealing with case reports.
Author Response
Thank you for your valuable comments.